# Enhancing Patient Recruitment Response in Clinical Trials: an Adaptive Learning Framework

**Xinying Fang**[1]                    **Shouhao Zhou**[1]

[1]Division of Biostatistics and Bioinformatics, Department of Public Health Sciences, Penn State University College of Medicine, Hershey, Pennsylvania, USA

## Abstract

Patient recruitment remains a key challenge in contemporary clinical trials, often leading to trial failures due to insufficient recruitment rates. To address this issue, we introduce a novel adaptive learning framework that integrates machine learning methods to facilitate evidence-informed recruitment. Through dynamic testing, predictive learning, and adaptive pruning of recruitment plans, the proposed framework ensures superiority over the conventional random assignment approach. We discuss the practical considerations for implementing this framework and conduct a simulation study to assess the overall response rates and chances of improvement. The findings suggest that the proposed approach can substantially enhance patient recruitment efficiency. By systematically optimizing recruitment plan allocation, this adaptive learning framework shows promise in addressing recruitment challenges across broad clinical research settings, potentially transforming how patient recruitment is managed in clinical trials.

## 1  INTRODUCTION

Patient recruitment is a principal challenge in conducting clinical trials [Friedman et al., 2015]. In a recent survey [eClinicalHealth], 86% of clinical trials did not meet enrolment timelines, and approximately one-third of phase III trials, representing the most rigid clinical studies that often take 5-15 years to implement and cost hundreds of millions of dollars, failed owing to participant enrolment problems. Our own experiences mirror these challenges, as seen in the PCORI-funded WISE trial, where slower-than-expected recruitment led to significant alterations [Sciamanna et al., 2018], including the modification of the study's primary endpoint and a reduction of total sample size by half.

Efforts have been undertaken to enhance patient recruitment in clinical trials. The recruitment guideline developed by the GREET project (Guidance to Recruitment: Examining Experiences at Clinical Trial Sites) [Zahren et al., 2021] identifies the availability of adequate staff resources, appropriate budget allocation, and proactive principal investigators as the top three facilitators of successful recruitment endeavors. However, it is essential to acknowledge that while these solutions demonstrate efficacy within specific trial contexts, their generalizability and efficiency are not guaranteed, subject to all kinds of predictable and completely unforeseen problems [Friedman et al., 2015].

Particularly, a more strict requirement for patient recruitment comes to pragmatic trials, such as studies to evaluate participants in the "SilverSneaker" program [Rovniak]. Since pragmatic trials are designed to assess the efficacy of interventions in real-world, routine practice conditions [Patsopoulos, 2011] and seek maximal heterogeneity in the clinical setting and patient characteristics, it requires a large sample size to give the intervention the best chance to demonstrate a beneficial effect [MOSIO]. Thus, with limited resources, efficient patient recruitment is vital to enhance generalizability for a wide range of participants.

Artificial Intelligence (AI) and Machine Learning (ML) have great potential in trial participant identification and selection: by using automated natural language processing tools, AI can effectively connect individuals to trials to increase participant identification [Miller et al., 2023, Weissler et al., 2021]; ML, particularly through neural network models, can reduce sample heterogeneity by identifying patients of specific characteristics with the prediction of benefit for patient selection [Harrer et al., 2019, Widera et al., 2023]. Despite these advancements, significant challenges persist, particularly in effectively encouraging participant responses to recruitment efforts. Even in a pool of well-identified potential participants, the recruitment response rate could be intolerably low (e.g., 3.2% in the WISE trial [Sciamanna et al., 2021], and 3% projected for Hispanic and Latino groups in the TIME trial [Sciamanna]), posing significant

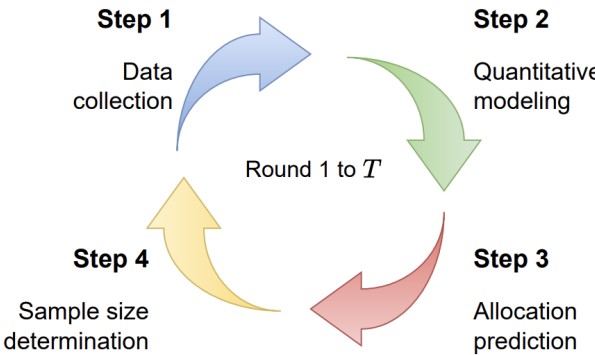

Figure 1: Concept graph of learning strategy within each round of recruitment.

difficulties in meeting enrollment targets and can jeopardize the success of a trial. Addressing this issue calls for the development and implementation of more effective and customized recruitment plans. However, due to the lack of specific data or evidence in individual trial contexts (e.g., various comparative interventions and targeted disease populations), it is still challenging to accurately predict the effectiveness of various recruitment plans, and estimate the achievable recruitment response rates. The need for innovative approaches that can navigate these complexities and effectively increase recruitment response rates is evident, making an important direction for further exploration and advancement using ML.

This paper seeks to transform clinical trial recruitment by harnessing ML to develop a cutting-edge, evidence-informed framework of adaptive recruitment strategy. Specifically, we will leverage predictive learning techniques over multiple candidate recruitment plans in a sequential recruitment setting. As shown in Figure 1, the process will go through $T$ rounds until the most effective recruitment plans are identified. Each round will go through four steps: data collection, quantitative modeling, recruitment plan allocation prediction, and sample size determination. By adaptively refining the selection of effective recruitment plans, we aim to achieve enhanced participant engagement by optimizing the recruitment plan allocation.

**Related works:** Sequential trial designs [Karrison et al., 2003, Li et al., 1995] adaptively update the weights of arm allocation to minimize the risk of inferior treatment assignments [Hu and Rosenberger, 2006]. The proposed recruitment strategy shares some similarities with sequential designs in adaptive learning, but they are distinct in the following aspects:

- *Study focus*: Sequential adaptive designs have an "arm-oriented" focus, aiming to identify the best treatment (arm) among the tested treatments. In contrast, the adaptive learning methods in our trial recruitment have a "response-oriented" focus. It aims to improve the overall recruitment response rate until little improvement can be made, irrespective of which recruitment plan (arm) achieves a good response.

- *Scale of interventions*: Traditional sequential adaptive designs can only handle a few treatments (small $K$) using classical statistical methods, whereas our approach is better suited for AI/ML techniques to systematically search within a large space of recruitment plans (large $K$).

- *Stage in clinical trials*: Sequential adaptive designs are implemented to allocate treatments during the intervention stage, while it is often costly and takes a few years to run even with a limited sample size. In contrast, the proposed adaptive learning framework targets the recruitment stage, which is fast-paced with a huge sample space (e.g., 175,000 in the SilverSneakers study).

Subsequently, the design methodologies required for these two types of studies differ significantly. While statistical approaches are often developed and applied for treatment assignment, ML methods are naturally suited to optimize efficiency in the recruitment setting. Our proposed approach establishes an adaptive ML framework to enhance participant recruitment in clinical trials. To the best of our knowledge, this innovative application of ML techniques to improve recruitment efficiency represents a groundbreaking development in the field.

## 2 AN ILLUSTRATIVE TRIAL EXAMPLE

A PCORI-funded clinical trial investigates the health and social effect of proactively utilizing the "SilverSneakers", an insurance-covered exercise program, among seniors with osteoarthritis [Rovniak]. Osteoarthritis is a common medical condition associated with pain, deterioration, and an increased risk of falls and fractures for the age group of 65 and over. In a pragmatic randomized parallel-group controlled trial setting, the randomization unit is the individual participant, who will be randomly assigned to one treatment arm (utilizing proactive care condition that provides support to activate insurance-funded SilverSneakers benefits) and one control arm (utilizing usual care condition that provided beneficiaries with their usual SilverSneakers benefits information) with a 1:1 ratio.

Scheduled for 2024-2025, this trial is budgeted to send out 175,000 recruitment letters for an enrollment target of 1,454 U.S. Medicare Advantage members. Despite SilverSneakers' substantial potential benefits and a large number of planned recruitment letters, the study still faces a significant challenge: recruiting enough participants. This concern is pivotal given the often lower recruitment response rates in disadvantaged groups, which makes participant recruitment a notably challenging task.

To enhance the recruitment process, a practical approach involves employing diverse recruitment modalities, such as utilizing different designs and features in recruitment letters to elicit higher response rates. These design features, each presented as a categorical variable with two or more levels, can be used individually or in combination, creating a high-dimensional sample space of recruitment (letter) plans.

In this vast sample space, it is critical to predict the most effective design features, or their combinations, to improve trial recruitment responses. Yet, pre-trial knowledge is scarce. Before the initiation of a trial, we have limited understanding of how potential participants might react to recruitment plans, due to the differences in proposed interventions, targeted study population, and specific trial context. In behavior research, experts' consensus may easily and significantly deviate from or contradict actual outcomes [Milkman et al., 2021]. This lack of foresight extends to predicting the effectiveness or ranking of recruitment plans.

Therefore, it illustrates an urgent need for new methodologies to enhance recruitment efficiency using adaptive strategies of learning and prediction. The integration of ML in sequential participant recruitment fills a gap in the existing literature, underlining the transformative potential for clinical trial breakthroughs in practice.

## 3 METHOD

The procedure of sequential participant recruitment aims to enhance the overall response rate (ORR) by optimizing the recruitment plan allocation. In this section, we first delineate the notations pertinent to the proposed approach, then delve into the modeling and design considerations essential for the selection of recruitment plan(s).

### 3.1 NOTATIONS

Suppose we have $K$ candidate recruitment plans to be distributed to $N$ potential participants. We assume that each participant can only receive one of the recruitment plans, at a random round $t \in T_0$ where $T_0$ is the planned maximum recruitment round, and the assignment is random following an allocation probability $w_k^{(t)}, k = 1, \ldots K$. Assume the $k$th recruitment plan has a true response rate of $p_k$, which is fixed but unknown, and can only be estimated from current trial data. We allow the adaptive recruitment process to stop early, so the actual total recruitment round $T \leq T_0$.

In the context of *SilverSneakers* trial with 8 binary design features, we specify $K = 2^8 = 256$ letter designs, $T_0 = 6$ maximum rounds, and $N = 175,000$ potential participants (i.e., each individual receives only one recruitment letter).

In a sequential recruitment process (e.g., Figure 2), the $N$ potential participants are randomly partitioned into $T$ sequential cohorts, and the individuals in the cohort $t$ will only

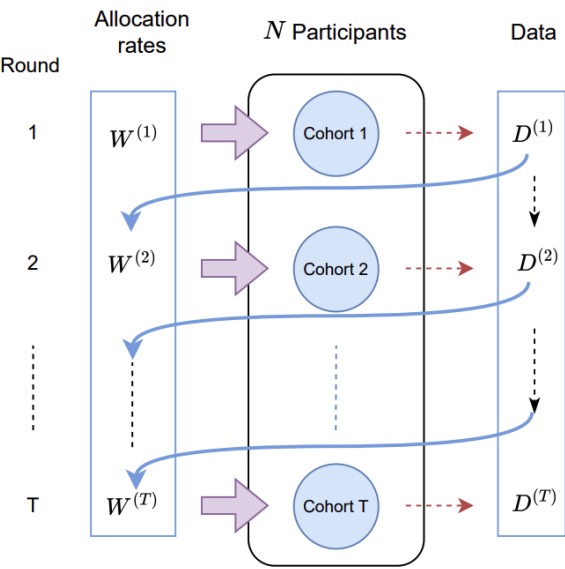

Figure 2: Adaptive procedure for recruitment plan assignment.

be reached out by the clinical team at the round $t$ of recruitment. The maximum number of patients involved at a non-terminal round (i.e., $t < T$) is $N/T_0$. The assignment probability $w_k^{(t)}$ for the $k$th recruitment plan could vary by time (the superscript in notation), possibly updated by the data $D^{(t-1)}$ collected up to previous $t - 1$ rounds of recruitment. If the adaptive learning approach is effective, we expect the assignment probabilities $w_k^{(t)}$ for high-performing recruitment plans (those with high response rates $p_k$) to increase over time. Conversely, the probabilities should decrease, potentially reaching zero, for underperforming recruitment plans with low response rates $p_k$. Overall, the algorithm will dynamically prioritize the more successful strategies while phasing out ineffective ones. For consistent notation, we denote $D^{(0)}$ as the prior data before initiating the recruitment. If there are no preliminary studies, $D^{(0)} = \varnothing$.

### 3.2 SEQUENTIAL RECRUITMENT PROCEDURE FOR ADAPTIVE LEARNING

Algorithm 1 presents the general recruitment strategy of an adaptive learning framework (Figure 2), which aims to allocate effective recruitment plans to improve the overall recruitment response rate. In the initial round ($t = 1$), all treatment plans are assigned equal probabilities $w_k^{(1)} = 1/K, k = 1, \cdots, K$, and participant responses are collected as $D^{(1)}$. For subsequent rounds $t > 1$, we follow the steps of Figure 1 to perform the adaptive allocation. Specifically, at round $t$, we have response data $D^{(t-1)}$ from previous rounds. An (ensemble) learning model is applied on $D^{(t-1)}$ to predict the response of recruitment plans $\hat{p}^{(t-1)}$. The allocation rates $W^{(t)}$ are derived from the predicted response rates $\hat{p}^{(t-1)}$, and we randomly assign the recruitment plans

to participants in cohort $t$ based on the allocation rates $\boldsymbol{W}^{(t)}$. New data $D^{(t)}/D^{(t-1)}$ are thus collected after potential participants respond to the assigned plans. This iterative process continues until the maximum number of rounds $T$ is reached or one of the early termination conditions 3(a)-(b) in Algorithm 1 is met.

Mathematically, the key step (i.e., step 3 in Figure 1) involves the determination of the cohort $t$-specific allocation rates $\boldsymbol{W}^{(t)} = (w_1^{(t)}, w_2^{(t)}, ..., w_K^{(t)})$, with

$$w_k^{(t)} \propto f_k(\hat{p}_1^{(t-1)}, ..., \hat{p}_K^{(t-1)}) \cdot g_k^{(t)}(\hat{p}_1^{(t-1)}, ..., \hat{p}_K^{(t-1)}),$$

where $f_k$ is some pre-specified randomization rule, and $g_k^{(t)}$ is an adaptive pruning factor that can be used to downweight recruitment plans that respond poorly, and $\sum_{k=1}^{K} w_k^{(t)} = 1$.

The learning performance may vary from the choices of $f_k$ and $g_k^{(t)}$, and yield different power and false discovery rate control. In the simulation study (Section 4), we test the adaptive learning performance when using the simple rule $f_k(\hat{p}_1^{(t-1)}, ..., \hat{p}_K^{(t-1)}) = \hat{p}_k^{(t-1)}$, proportional to the predicted response rate. Additionally, we assign the adaptive pruning factor $g_k^{(t)}$ at value 1 to a cluster of recruitment plans with the highest predicted response rates in round $t$, denoted by $C^{(t)}$. All the recruitment plan $k \notin C^{(t)}$ are pruned with $g_k^{(t)} = 0$. By applying the K-means method [Lloyd, 1982, MacQueen, 1967], this selected recruitment plan set $C^{(t)}$ is determined out of the previously selected $C^{(t-1)}$ in round $t-1$ to satisfy the monotonic condition. The silhouette score [Rousseeuw, 1987] is used to select the best number of clusters. If it demonstrates the effectiveness of recruitment in this simple setting, we expect the adaptive learning performance to be further enhanced with tailored ML methods in real data applications.

We now discuss the theoretical properties in this setting.

**Lemma 1.** For any rule $f_k(p_1, p_2, \cdots, p_K) \propto p_k$,

$$\sum_k w_k p_k = \sum_k f_k p_k / \sum_k f_k \geq \sum_k p_k / K.$$

The derivation of Lemma 1 employs the Cauchy–Schwarz inequality for its proof (Supplementary Section A.1), and the equality holds *iff* $p_1 = p_2 = \cdots = p_K$. Incorporating the law of large numbers leads to the subsequent remark, suggesting that it is always safe to apply a consistent learning strategy in recruitment:

**Remark 1.** (*Non-inferiority*) If a learning method yields recruitment response estimators that converge consistently (i.e., $\lim_{n\to\infty} \hat{p}_k = p_k$, for $k = 1, \ldots, K$), then a recruitment strategy based on $f_k(\hat{p}_1, \hat{p}_2, \ldots, \hat{p}_K) \propto \hat{p}_k$ will be statistically non-inferior to the conventional strategy that assigns recruitment equally (i.e., $f_k(\hat{p}_1, \hat{p}_2, \ldots, \hat{p}_K) = 1/K$), with probability 1.

Additionally, we have the following property of adaptive recruitment plan selection, given a pruning factor $g_k^{(t)} \in \{0, 1\}$ on $t \in [1, T]$, which satisfies the boundary constraints

$$g_k^{(1)} = 1, \text{ for } k = 1, \ldots, K$$
$$\sum_{k=1}^{K} g_k^{(T)} \geq 1,$$

and, to consistently exclude the less effective recruitment plans among the preceding round, imposing the monotonic condition

$$g_k^{(t)} \leq g_k^{(t-1)},$$
$$\sum_{k=1}^{K} g_k^{(t)} < \sum_{k=1}^{K} g_k^{(t-1)}, \text{ for } 2 \leq t \leq T :$$

**Lemma 2.** (*Optimality*) Without loss of generality, we assume that the true recruitment response rate $1 \geq p_1 > p_2 > \cdots > p_K \geq 0$. For any consistent rule $f_k(p_1, p_2, \cdots, p_K) \propto p_k$, if combined with a strict pruning factor $g_k^{(t)}(p_1, p_2, \cdots, p_K)$ satisfying

$$g_k^{(T)}(p_1, p_2, \cdots, p_K) = \begin{cases} 1, & k = \text{argmax}_k p_k = 1 \\ 0, & k \neq 1 \end{cases},$$

we have

$$\sum_k w_k^{(t)} p_k = \sum_k f_k g_k^{(t)} p_k / \sum_k f_k g_k^{(t)}$$
$$\leq \sum_k f_k g_k^{(T)} p_k / \sum_k f_k g_k^{(T)} = p_1.$$

The proof is included in Supplementary Section A.2. Lemma 2 suggests that, when the recruitment plans can be completely ranked in terms of the recruitment response rates, the optimal response rate will be achieved, at least in the last cohort, by adopting the strict pruning factor to select the most effective recruitment plan. This interesting result can also be easily generalized to any semi-strict pruning factor $g_k^{(t)}(p_1, p_2, \cdots, p_K)$ with

$$g_k^{(T)}(p_1, p_2, \cdots, p_K) = \begin{cases} 1, & k \in \{1, \ldots, K_0\} \\ 0, & k \in \{K_0 + 1, \ldots, K\} \end{cases},$$

for the best subset with $K_0$ recruitment plans, to obtain

$$\sum_k w_k^{(t)} p_k \geq \sum_k w_k^{(t-1)} p_k, \text{ for } 2 \leq t \leq T$$
$$\text{and } \sum_k w_k^{(T)} p_k \geq p_{K_0}.$$

This result is important because, in practice, we don't expect a large $T$ for many rounds of learning to guarantee the identification of the most effective recruitment plan; nonetheless, it is still promising to adopt an adaptive pruning process

**Algorithm 1:** Procedure for Adaptive Learning Framework

---

**Inputs:** initial round $t = 1$, total round $T = T_0$, sample size for round 1 $n^{(1)} = N/T_0$.

**while** $t \leq T$ **do**

    **if** *t = 1* **then**

        | Randomly assign all patients in cohort 1 according to $w_k^{(1)} = 1/K$, where $k = 1, ..., K$, and obtain data $D^{(1)}$;

    **else**

        1. Given the data $D^{(t-1)}$ collected up to round $t - 1$, we apply the ensemble model to predict the plan response rates $\hat{\boldsymbol{p}}^{(t-1)} = (\hat{p}_1^{(t-1)}, \hat{p}_2^{(t-1)}, ..., \hat{p}_K^{(t-1)})$;

        2. Calculate the allocation rates $\boldsymbol{W}^{(t)} = (w_1^{(t)}, w_2^{(t)}, ..., w_K^{(t)})$ with

$$w_k^{(t)} \propto f_k(\hat{p}_1^{(t-1)}, \hat{p}_2^{(t-1)}, ..., \hat{p}_K^{(t-1)}) \cdot g_k^{(t)}(\hat{p}_1^{(t-1)}, \hat{p}_2^{(t-1)}, ..., \hat{p}_K^{(t-1)})$$

        based on pre-specified randomization rule $f_k$, pruning factor $g_k^{(t)}$, and $\sum_{k=1}^{K} w_k^{(t)} = 1$;

        3. **if** $t < T$ **then if**

            (a) $n_{\min}^{(t)} < 0$ (the precision of the observed ORR has met the power requirement);
            **or**

            (b) $\exists\, k, w_k^{(t)} = 1$ (single recruitment plan selected for next cohort sampling);
            **or**

            (c) $\widehat{ORR}^{(t)} - \widehat{ORR}^{(t-1)} < \epsilon$ (limited improvement on predicted ORR);

           **then** (Early stopping)

                Terminate the adaptive learning with $T = t$ by combining all the rest samples into a single cohort with sample size $n^{(t)} = N - \sum_{s=1}^{t-1} n^{(s)}$;

           **else**

                Calculate cohort $t$ sample size, $n^{(t)}$ (Section 3.3.2);

        4. Randomly assign recruitment plans $1, ..., K$ to individuals in cohort $t$ according to $\boldsymbol{W}^{(t)}$ and collect response data, which will be combined with data $D^{(t-1)}$ collected in previous rounds to generate the updated data $D^{(t)}$;

        5. $t = t + 1$;

    **end**

**end**

**Result:** Participants response data collected up to round $T$, $D^{(T)}$, and overall response rate over $N$ samples, $ORR^{(T)}$.

---

to exclude some ineffective allocation plans and improve the recruitment responses. The K-means derived $g_k^{(t)}$ is a typical example.

When the sample size is large enough to conduct consistent response rate estimation, the subsequent proposition holds following the Remark 1,

**Proposition 1.** (*Superiority*) Jointly with a pruning strategy $g_k^{(t)}(\hat{p}_1, \hat{p}_2, \ldots, \hat{p}_K)$ in patient allocation, the adaptive learning strategy $w_k^{(t)} \propto f_k \cdot g_k^{(t)}$ can consistently improve recruitment efficiency over time, if some $p_k$'s are not equal.

## 3.3 MODELING AND DESIGN CONSIDERATIONS IN SELECTING THE RECRUITMENT PLAN(S)

Careful considerations should be taken in the above adaptive learning framework. Below we demonstrate a feasible approach and its justifications, specifically regarding the ensemble modeling, total round determination, early termination rules, and sample size calculation for each round.

### 3.3.1 Ensemble modeling for response rate prediction

We illustrate the response prediction model using ensemble learning, which combines the predictions from multiple ML algorithms (as base learners) to make robust predictions [Dietterich, 2000, Guzman et al., 2015, da Silva et al., 2014]. The ensemble model can be fine-tuned using the best-fitting parameters, which are identified through a grid search method coupled with 10-fold cross-validation. The 7 selected ML algorithms for simulation study are categorized into two groups: parametric models (including logistic regression [Cox, 1958], lasso regression [Tibshirani, 1996], ridge regression [Hoerl and Kennard, 2000], and non-parametric models, such as gradient boosting ma-

chine (GBM) [Friedman, 2001], random forest (RF) [Ho, 1995], Extreme Gradient Boosting (XGBoost) [Chen and Guestrin, 2016], and artificial neural networks (NNs) [Grossberg, 1988]. In general, the selection of the base learners can be customized, depending on the study objective, for different datasets.

### 3.3.2 Sample size calculation

When the total sample size is limited, we can determine the minimum sample size required for round $t$ by considering the observed response rate for round $t-1$, denoted as $\hat{p}^{(t-1)}$, and an arbitrary expected effect size improvement $\Delta$. Assuming a target power of $1 - \beta^{(t)}$, where $\beta^{(t)}$ represents the Type II error rate at round $t$, we conduct hypothesis testing with the following hypotheses:

$$H_0 : p^{(t)} - p^{(t-1)} = 0$$
$$H_1 : p^{(t)} - p^{(t-1)} > 0$$

The minimum sample size required to reject the null hypothesis at round t, denoted as $n_{min}^{(t)}$, is:

$$n_{min}^{(t)} = \frac{\hat{p}^{(t)}(1-\hat{p}^{(t)})}{\frac{\Delta^2}{(Z_{1-\alpha}+Z_{1-\beta^{(t)}})^2} - \frac{\hat{p}^{(t-1)}(1-\hat{p}^{(t-1)})}{n^{(t-1)}}}$$

Here, $n^{(t-1)}$ represents the observed sample size for round $t-1$, and $\alpha$ denotes the Type I error rate. $\hat{p}^{(t)}$ is calculated as $\hat{p}^{(t)} = \hat{p}^{(t-1)} + \Delta$. $Z_{1-\alpha}$ and $Z_{1-\beta^{(t)}}$ are critical values from the standard normal distribution. Additionally, the minimum sample size is constrained by the total sample size divided by the number of rounds, i.e., $N/T$. Hence, the final minimum sample size is determined as $n^{(t)} = \min\{n_{min}^{(t)}, N/T\}$.

### 3.3.3 Early termination

The adaptive learning procedure may stop early under three conditions that no significant improvement in recruitment allocation can be further made through adaptive learning. Firstly, if the precision of the observed ORR has met the power requirement, indicated by $n_{min}^{(t)} < 0$. Secondly, if only one recruitment plan remains in $D^{(t-1)}$. Thirdly, the process halts early when the following condition is met:

$$\widehat{ORR}^{(t)} - \widehat{ORR}^{(t-1)} < \epsilon$$

where $\widehat{ORR}^{(t)}$ represents the predicted ORR across all recruitment plans using data $D^{(t-1)}$. According to the adaptive learning framework (Section 3.2), recruitment plans with higher response rates are prioritized and assigned greater weight for subsequent rounds of the trial. Based on the definition of $\widehat{ORR}^{(t-1)}$ and $\widehat{ORR}^{(t)}$, we can readily establish the following proposition:

**Proposition 2.** $\widehat{ORR}^{(t)} - \widehat{ORR}^{(t-1)} > 0$

The third stopping criterion signifies that the updated recruitment plans at round $t$ yield only marginal enhancements to the overall response rate. When any of the three conditions is met, the adaptive learning terminates at $T = t$ by consolidating all remaining samples into a single cohort with an allocation probability of $W^{(T)} = W^{(t)}$.

### 3.3.4 Total round determination

For adaptive learning, it is also important to specify the total round $T$, which could be pre-fixed or considered random. From a cost-effective perspective, the total round $T$ is constrained by the study recruitment duration ($Tot_t$) and total cost ($Tot_c$). We denote $C_1$ the fixed cost for each round of recruitment, $C_2$ the cost for each letter or sample size, and $Time_r$ the projected duration for each round. With $N$ potential participants to be reached out in total $T$ rounds, the estimated total cost and time duration for the recruitment will be:

$$\widehat{Cost} = T \times C_1 + N \times C_2$$
$$\widehat{Time} = T \times Time_r$$

Therefore, the total round $T$ is determined by the two conditions: $\widehat{Cost} \leq Tot_c$ and $\widehat{Time} \leq Tot_t$. For feasibility, we fixed the total round to be 6 in the simulation (Section 4).

In practice, researchers should begin with a conservative estimate of the time frame required for each round of patient recruitment and response collection, denoted as $Time_r$ (e.g., 2 months). Given the total scheduled duration for the recruitment phase, $Tot_t$ (e.g., 1 year), the maximum number of rounds $T_{max}$ can be determined as $T_{max} = Tot_t/Time_r$. The total recruitment cost should then be evaluated for $T_{max}$ rounds to ensure it meets the cost constraints. If the cost exceeds the available resources, the number of rounds $T$ can be reduced below $T_{max}$ accordingly.

To assess the potential impact of varying $T$, simulation studies will be conducted as sensitivity analyses, examining the effects of increasing or decreasing the number of rounds. Generally, we recommend $T = 4$ to $8$ rounds for effective adaptive learning, as this range tends to strike a balance between computational efficiency and the ability to learn and adapt over multiple iterations. However, the final selection of $T$ should be made jointly with the study's principal investigator (PI), considering the overall recruitment strategy and resource limitations.

### 3.3.5 Total recruitment plan determination

Assume that we have determined the total round $T$, an estimated response rate ($\theta_0$), and an estimated minimum number of responders for each recruitment plan ($S$), this leads us to the inequality that $K * T * S < N * \theta_0$, where $K$ signifies

the total number of recruitment plans and $N$ represents the total sample size. The inequality suggests that $K$ is subject to the constraint:

$$K < \frac{N * \theta_0}{T * S}$$

This constraint ensures the adequacy of data for our machine learning model to effectively discern the impact of different recruitment plans. By adhering to this constraint, we guarantee the availability of sufficient data points necessary for accurate analysis and interpretation of the effects of the recruitment strategies.

## 4 SIMULATION

We conduct simulation studies and design three scenarios to examine the efficiency of adaptive ensemble learning for participant recruitment in the context of the SilverSneakers program (Section 2). Details on the simulation settings are described in the Supplementary Section B.

The simulation results (Tables 1 and Tables 2 - 6 in the Supplementary Material) highlight the progressive nature of adaptive learning, wherein recruitment plans with superior response rates are increasingly favored over successive rounds. The ORRs of the last round closely approximate the highest true response rate (RR) and the ORR converges notably to the highest true RR starting after round 2 (e.g., Table 1(A)). The number of remaining recruitment plans in the final round is around 2 (e.g., Table 1(B)), which demonstrates that the adaptive learning framework discards ineffective recruitment plans with zero weight at the first two rounds.

Notably, the early stopping rate is high (e.g., Table 1(C)) and

mostly is due to the single recruitment plan selected for the next cohort sampling (Algorithm 1). The sample size used for the last round for Scenarios 1 and 2 is at least 110,000, which is at least 62.8% of the total 175,000 sample size. The adaptive learning approach achieves its objectives using less than half of the available data, underscoring that the approach is efficient in identifying and selecting the most promising recruitment plan.

The true response rate assignment in Scenario 1 does not favor any design features. As depicted in Tables 1(A) and 2(A)), while tree-based methods and the two ensemble learning approaches manage to attain the highest true response rate (0.097, (0.003)), logistic ridge regression falls short with an overall ORR of 0.080 (0.010) and adaptive learning ORR of 0.087 (0.012). Despite the inclusion of logistic ridge regression within the ensemble learning methods, the robustness inherent in ensemble learning enables them to maintain performance levels on par with tree-based methods, random forest, and XGBoost. This signifies the efficacy of ensemble learning in mitigating the risk associated with selecting less effective machine learning methods.

Scenario 2(a) constructs an additive setting for design features, which favors the logistic regression. Consequently, we observe superior performance of logistic ridge regression compared to random forest and XGBoost, as delineated in Tables 3 and 4. Conversely, Scenario 2(b) adds an interaction term for design features, facilitating tree-based methods to outperform logistic ridge regression, as evident in Tables 5 and 6. Although these two settings are designed to favor different base learners, the ensemble learning methods can still take advantage of containing at least one of the favored

Table 1: Simulation results for Scenario 1 with 5 design features. Values for ORRs, plan numbers, expected rounds, and sample size for last round are mean with standard deviation in the parenthesis.

(A)

| Scenario 1 (5 designs) | Round 1 ORR[(1)] | Round 2 ORR[(2)] | Round 3 ORR[(3)] | Round 4 ORR[(4)] | Round 5 ORR[(5)] | Overall ORR | Adaptive Learning ORR | Highest True RR |
|---|---|---|---|---|---|---|---|---|
| Logistic ridge regression | 0.055 (0.005) | 0.067 (0.011) | 0.080 (0.013) | 0.088 (0.008) | 0.092 (0.006) | 0.080 (0.010) | 0.087 (0.012) | 0.097 (0.003) |
| Random forest | 0.055 (0.005) | 0.086 (0.008) | 0.093 (0.005) | 0.095 (0.004) | 0.096 (0.003) | 0.087 (0.003) | 0.096 (0.004) | 0.097 (0.003) |
| XGBoost | 0.055 (0.005) | 0.086 (0.008) | 0.093 (0.005) | 0.095 (0.005) | 0.096 (0.003) | 0.087 (0.003) | 0.096 (0.003) | 0.097 (0.003) |
| Ensemble learning - 3 learners | 0.055 (0.005) | 0.085 (0.009) | 0.092 (0.005) | 0.095 (0.004) | 0.096 (0.004) | 0.087 (0.003) | 0.096 (0.003) | 0.097 (0.003) |
| Ensemble learning - 7 learners | 0.055 (0.005) | 0.084 (0.008) | 0.092 (0.005) | 0.095 (0.004) | 0.096 (0.004) | 0.087 (0.003) | 0.095 (0.004) | 0.097 (0.003) |

(B)

| Scenario 1 (5 designs) | Round 1 Plan number | Round 2 Plan number | Round 3 Plan number | Round 4 Plan number | Round 5 Plan number | Expected rounds |
|---|---|---|---|---|---|---|
| Logistic ridge regression | 32 | 9.5 (8.0) | 3.2 (3.0) | 2.0 (1.4) | 1.5 (0.7) | 3.8 (0.9) |
| Random forest | 32 | 9.8 (6.6) | 4.0 (2.9) | 2.4 (1.8) | 1.8 (1.2) | 4.2 (0.9) |
| XGBoost | 32 | 9.3 (6.2) | 3.7 (2.9) | 2.4 (2.0) | 1.7 (1.1) | 4.0 (0.9) |
| Ensemble learning - 3 learners | 32 | 10.2 (6.6) | 3.9 (3.0) | 2.3 (1.6) | 1.7 (1.3) | 4.2 (0.8) |
| Ensemble learning - 7 learners | 32 | 8.6 (5.8) | 3.4 (2.5) | 2.1 (1.5) | 1.5 (1.0) | 4.0 (0.9) |

(C)

| Scenario 1 (5 designs) | Early Stopping | Due to plan number | Due to ORR restriction | Due to sample size | Sample size for last round | Better performance |
|---|---|---|---|---|---|---|
| Logistic ridge regression | 73% | 73% | 0% | 0% | 130411 (5760) | 99% |
| Random forest | 55% | 55% | 0% | 0% | 125088 (7340) | 100% |
| XGBoost | 63% | 63% | 0% | 0% | 126751 (7443) | 100% |
| Ensemble learning - 3 learners | 55% | 55% | 0% | 0% | 125433 (7209) | 100% |
| Ensemble learning - 7 learners | 63% | 63% | 0% | 0% | 126510 (7489) | 100% |

learners to perform comparably well. Moreover, comparing ensemble methods with 3 base learners to 7 base learners reveals that incorporating more methods does not compromise overall performance and can enhance it in certain instances. For instance (Table 4(A)), the ensemble learning with 7 learners outperforms other methods, including the ensemble learning with 3 learners from round 2, while the latter catches up with the performance of the 7-learner ensemble learning from round 3. This observation underscores the robustness of ensemble learning across diverse conditions and further highlights its adaptability and effectiveness in various scenarios.

We also conducted a comparison between the proposed adaptive learning framework and the benchmark in three scenarios. Since there is no recruitment plan selection and allocation prediction, the ORR of the benchmark is 0.055 all the time. However, the examination of the results tables reveals that the overall ORRs of all candidate methods in Scenarios 1 and 2 closely approximate the highest true response rates. Specifically, these rates are 0.097 and 0.1 for Scenario 1 with 5 and 8 design features, and 0.068 for Scenario 2. The chances of better performance against the benchmark in Table (C) of all results tables are almost 100% of the time for the ensemble learning methods. In Scenario 3, due to the absence of recruitment plan selection and plan allocation rate predictions, both the benchmark and the adaptive learning approach maintain a steady ORR of 0.055 throughout the entire process, indicating a non-inferior performance of the adaptive learning approach. These findings indicate that the adaptive learning framework exhibits performance comparable to, or notably superior to, the random approach (benchmark). This suggests the robustness and effectiveness of the adaptive learning approach in optimizing recruitment plans under various conditions, thus affirming its suitability for practical implementation.

In summary, we designed three scenarios to demonstrate the robustness of the adaptive learning framework with the ensemble learning method. Compared with the random approach, the adaptive learning framework can effectively select the most effective recruitment plan in a fast and efficient manner. The incorporation of ensemble learning into allocation prediction mitigates the risk of choosing undesirable machine learning methods, ensuring consistent and robust performance across diverse scenarios.

## 5  EXTENSIONS

Below we highlight some potential extensions of the illustrated method.

Refining pruning strategy: In pursuit of a more streamlined and effective recruitment plan selection process, K-means could be replaced with more effective approaches, such as X-means [Pelleg and Moore, 2000]. This transition enables dynamic selection in the optimal number of clusters, over-coming the limitations associated with K-means. Additionally, X-means enhances computational efficiency, making it a more robust choice in applications.

Global optimization in response estimation: The proposed approach focuses on maximizing predicted plan response rates. Alternative methods, such as the multi-armed bandit approach [Agrawal and Goyal, 2012, Oh and Iyengar, 2019], which simultaneously balances exploration with exploitation, may lead to a more efficient adaptive recruitment strategy by globally maximizing the total predictive reward.

Enhancing recruitment for underrepresented patients: A group-specific recruitment allocation plan can be implemented to mitigate health disparities and enhance the recruitment of underrepresented populations. This extension involves clustering patients based on their demographic and clinical characteristics and optimizing recruitment strategies tailored to each group.

Incorporating external evidence: When relevant external data or experts' opinions are available, they can be converted as the prior distributions and the allocation optimization could also be conducted using the Bayes learning [Gelman et al., 2013]. Furthermore, in clinically diverse settings with multiple patient groups, it is compelling to employ learning methods that can incorporate both plan features and patient characteristics. Potentially, it will lead to better identification of optimal recruitment plans for specific patient subgroups, opening a door to diverse efficiency gains.

Sequential assignment for non-responders: In the context of rare diseases where the pool of eligible patients is limited, the proposed approach can be adapted to re-assign non-responders to receive additional recruitment plans. This extension tests ML-guided recruitment strategies in sequential order for patients who do not respond, maximizing the chances of successful enrollment in the study.

## 6  CONCLUSION

Patient recruitment remains a critical challenge in large-scale pragmatic clinical trials, necessitating extensive sample sizes and diverse patient populations. Traditional strategies often struggle to meet demanding recruitment requirements across varied clinical settings. We proposed a novel adaptive learning framework integrating ensemble learning to iteratively optimize patient recruitment. Through simulations, we demonstrated the proposed framework could efficiently identify and prioritize the most effective recruitment plans while mitigating the risk of selecting suboptimal recruitment plans. This work establishes a foundation for leveraging AI/ML to address longstanding recruitment challenges, facilitating more efficient pragmatic trials by substantially improving recruitment rates and accelerating clinical research.

## Author Contributions

List of Authors: Xinying Fang (X.F.), Shouhao Zhou (S.Z.)

S.Z. conceived the original idea. X.F. created the code, ran the experiments, and generated the figures, with regular feedback from S.Z. X.F. and S.Z. wrote the paper.

## Acknowledgements

We would like to thank the editor and five anonymous reviewers for their insightful comments. We are grateful to Drs. Liza Rovniak and Christopher Sciamanna for discussion of the illustrative study.

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

# Supplementary Material

**Xinying Fang**[1]                          **Shouhao Zhou**[1]

[1]Division of Biostatistics and Bioinformatics, Department of Public Health Sciences, Penn State University College of Medicine, Hershey, Pennsylvania, USA

## A  PROOFS

### A.1  PROOF OF LEMMA 1

According to the Cauchy-Schwarz inequality, we can get

$$\left(\sum_k f_k * 1\right)^2 \leq K \sum_k f_k^2$$

The inequality is rearranged to

$$\sum_k f_k^2 / \sum_k f_k \geq \sum_k f_k / K$$

Thus, with $f_k(p_1, p_2, \cdots, p_K) \propto p_k$, we can get

$$\sum_k f_k p_k / \sum_k f_k \geq \sum p_k / K.$$

### A.2  PROOF OF LEMMA 2

For $1 \geq p_1 \geq p_2 \geq \cdots > p_K \geq 0$, we have

$$\begin{aligned}
\sum_k w_k^{(t)} p_k &= \sum_k f_k g_k^{(t)} p_k / \sum_k f_k g_k^{(t)} \\
&\leq \sum_k f_k g_k^{(t)} p_1 / \sum_k f_k g_k^{(t)} = p_1
\end{aligned}$$

Also, we know that $g_k^{(T)} = 1$ if $k = 1$, so

$$\sum_k f_k g_k^{(T)} p_k / \sum_k f_k g_k^{(T)} = f_1 p_1 / f_1 = p_1.$$

Therefore, we can conclude that

$$\begin{aligned}
\sum_k w_k^{(t)} p_k &= \sum_k f_k g_k^{(t)} p_k / \sum_k f_k g_k^{(t)} \\
&\leq \sum_k f_k g_k^{(T)} p_k / \sum_k f_k g_k^{(T)} = p_1.
\end{aligned}$$

# B SIMULATION SETTING

For an illustrative purpose, here we simplify the setting by disregarding participant-specific characteristics and test on 5 and 8 binary design features, each leading to $2^5 = 32$ and $2^8 = 256$ candidate recruitment plans, respectively. We consider five methods for comparison, including logistic regression with l2 penalty, random forest, XGBoost, ensemble learning with these three methods, and ensemble learning with the seven methods mentioned in Section 3.3.1.

The true underlying response rate $p_k$ of each recruitment plan is defined using the following three scenarios:

1. The true response rates for each recruitment plan are randomly assigned within $[0.01, 0.1]$.

2. Logistic regression scenario:

   (a) Assign fixed coefficients to recruitment plans. Let $\beta_1$ for design feature 1 be 0.5, $\beta_2$ for design feature 2 be -0.5, and coefficients for all other designs be 0. Then, the response rates for each recruitment plan is $inv.logit(\beta_1 x_1 + \beta_2 x_2) * 0.11$, where $x_1$ and $x_2$ are binary indicators for design features 1 and 2. The multiplying factor $0.11$ is applied to maintain an expected overall recruitment response rate to be 0.055.

   (b) Assign fixed coefficients along with an interaction. We assign $\beta_1 = -0.5$ for design feature 1 and $\beta_{12} = 1$ for the interaction between design features 1 and 2. The coefficients for all other designs are 0. Then, the response rates for each recruitment plan is $inv.logit(\beta_1 x_1 + \beta_{12} x_1 x_2) * 0.11$. The multiplying factor $0.11$ is applied to maintain an expected overall recruitment response rate to be 0.055.

3. The true response rates for each recruitment plan are equal to 0.055.

While all scenarios have the same expected overall recruitment response rates of 0.055, Scenario 3 is the worst-case scenario for learning when no improvement could be made. Nevertheless, we include it to examine the non-inferiority of the proposed learning procedure.

We limit the total sample size (letters) to 175,000 and the total rounds of the experiment, $T$, are 5 and 6 for 5 and 8 design features, respectively. Thus, approximately $175000/5 \approx 35000$ and $175000/6 \approx 29167$ participants will be reached out as cohort 1 at round 1. The remaining samples will be allocated across subsequent rounds based on the sample size calculation (Section 3.3.2) and the early termination rule (Section 3.3.3). We repeat the data-generating process 100 times, to calculate the ORR within each cohort and over the whole sample. As a benchmark, we employ a random approach where recruitment plans are randomly assigned to participants at each round. To evaluate the performance of the adaptive learning framework against this benchmark, we conduct binomial hypothesis testing at each replication to determine the overall chances of better performance against the benchmark by the adaptive learning approach. The adaptive learning procedure for the simulation study are illustrated in Algorithm 2.

---

**Algorithm 2:** Adaptive learning procedure for the simulation study

---

**Inputs:** initial round $t = 1$, total round $T = T_0$, sample size for round 1 $n^{(1)} = N/T_0$.

**while** $t \leq T$ **do**

  **if** $t = 1$ **then**

      Randomly assign all patients in cohort 1 according to $w_k^{(1)} = 1/K$, where $k = 1, ..., K$, and obtain data $D^{(1)}$;

  **else**

    1. Given the data $D^{(t-1)}$ collected up to round $t-1$,

        (a) Apply a learning model (e.g., an ensemble model or a base learner) to predict the plan response rates $\hat{p}_k^{(t-1)}$ among the recruitment plans $k \in C^{(t-1)}$ (i.e., the set of recruitment plans with the adaptive pruning factor $g_k^{(t-1)} = 1$);

        (b) Perform K-means clustering on the predicted plan response rates $\{\hat{p}_k^{(t-1)}\}$, $k \in C^{(t-1)}$;

        (c) Assign (keep) $g_k^{(t)} = 1$ to the recruitment plans in the best-performed cluster, denoted by $C^{(t)}$. All the other recruitment plans $k \notin C^{(t)}$ are pruned with $g_k^{(t)} = 0$;

    2. Calculate the allocation rates $\boldsymbol{W}^{(t)} = (w_1^{(t)}, w_2^{(t)}, ..., w_K^{(t)})$ with

$$w_k^{(t)} = \frac{\hat{p}_k^{(t-1)} \cdot g_k^{(t)}}{\sum_k \hat{p}_k^{t-1} \cdot g_k^{(t)}}$$

    3. **if** $t < T$ **then if**

        (a) $n_{\min}^{(t)} < 0$ (the precision of the observed ORR has met the power requirement);

        **or**

        (b) $\exists\, k,\, w_k^{(t)} = 1$ (single recruitment plan selected for next cohort sampling);

        **or**

        (c) $\widehat{ORR}^{(t)} - \widehat{ORR}^{(t-1)} < \epsilon$ (limited improvement on predicted ORR);

      **then** (Early stopping)

        Terminate the adaptive learning with $T = t$ by combining all the rest samples into a single cohort with sample size $n^{(t)} = N - \sum_{s=1}^{t-1} n^{(s)}$;

      **else**

        Calculate cohort $t$ sample size, $n^{(t)}$ (Section 3.3.2);

    4. Randomly assign recruitment plans $1, ..., K$ to individuals in cohort $t$ according to $\boldsymbol{W}^{(t)}$ and collect response data, which will be combined with data $D^{(t-1)}$ collected in previous rounds to generate the updated data $D^{(t)}$;

    5. $t = t + 1$;

  **end**

**end**

**Result:** Participants response data collected up to round $T$, $D^{(T)}$, and overall response rate over $N$ samples, $ORR^{(T)}$.

---

# C  ADDITIONAL SIMULATION RESULTS

Table 2: Simulation results for Scenario 1 with 8 design features. Values for ORRs, plan numbers, expected rounds, and sample size for last round are mean with standard deviation in the parenthesis.

(A)

| Scenario 1 (8 design features) | Round 1 ORR[(1)] | Round 2 ORR[(2)] | Round 3 ORR[(3)] | Round 4 ORR[(4)] | Round 5 ORR[(5)] | Round 6 ORR[(6)] | Overall ORR | Adaptive Learning ORR | Highest True RR |
|---|---|---|---|---|---|---|---|---|---|
| Logistic ridge regression | 0.055 (0.002) | 0.059 (0.003) | 0.066 (0.008) | 0.075 (0.010) | 0.083 (0.011) | 0.089 (0.009) | 0.080 (0.007) | 0.085 (0.009) | 0.1 (0.000) |
| Random forest | 0.055 (0.002) | 0.081 (0.007) | 0.089 (0.007) | 0.093 (0.005) | 0.096 (0.004) | 0.097 (0.002) | 0.089 (0.002) | 0.096 (0.003) | 0.1 (0.000) |
| XGBoost | 0.055 (0.002) | 0.079 (0.008) | 0.088 (0.007) | 0.093 (0.005) | 0.096 (0.003) | 0.097 (0.002) | 0.089 (0.002) | 0.096 (0.003) | 0.1 (0.000) |
| Ensemble learning - 3 learners | 0.055 (0.002) | 0.079 (0.008) | 0.089 (0.006) | 0.093 (0.005) | 0.096 (0.004) | 0.097 (0.003) | 0.089 (0.002) | 0.096 (0.003) | 0.1 (0.000) |
| Ensemble learning - 7 learners | 0.055 (0.002) | 0.078 (0.009) | 0.088 (0.006) | 0.093 (0.005) | 0.096 (0.004) | 0.097 (0.004) | 0.088 (0.003) | 0.099 (0.004) | 0.1 (0.000) |

(B)

| Scenario 1 (8 design features) | Round 1 Plan number | Round 2 Plan number | Round 3 Plan number | Round 4 Plan number | Round 5 Plan number | Round 6 Plan number | Expected rounds |
|---|---|---|---|---|---|---|---|
| Logistic ridge regression | 256 | 97.5 (58.0) | 32.8 (32.8) | 12.3 (19.4) | 4.8 (6.5) | 2.4 (2.1) | 5.5 (0.8) |
| Random forest | 256 | 99.6 (61.7) | 38.1 (43.2) | 10.9 (16.5) | 4.6 (4.6) | 2.6 (2.3) | 5.3 (0.9) |
| XGBoost | 256 | 97.2 (60.1) | 39.1 (39.5) | 11.4 (15.3) | 4.6 (9.1) | 2.1 (1.7) | 5.5 (0.7) |
| Ensemble learning - 3 learners | 256 | 102.6 (58.4) | 34.0 (35.8) | 11.1 (11.6) | 4.6 (5.5) | 2.5 (3.2) | 5.5 (0.8) |
| Ensemble learning - 7 learners | 256 | 95.0 (58.4) | 26.7 (31.2) | 9.2 (13.7) | 3.6 (5.3) | 2.4 (2.5) | 5.2 (0.9) |

(C)

| Scenario 1 (8 design features) | Early Stopping | Due to plan number | Due to ORR restriction | Due to sample size | Sample size for last round | Better performance |
|---|---|---|---|---|---|---|
| Logistic ridge regression | 34% | 31% | 0% | 3% | 123916 (7556) | 100% |
| Random forest | 43% | 42% | 0% | 1% | 118781 (9656) | 100% |
| XGBoost | 37% | 35% | 0% | 2% | 116989 (9351) | 100% |
| Ensemble learning - 3 learners | 30% | 27% | 0% | 3% | 117005 (9722) | 100% |
| Ensemble learning - 7 learners | 49% | 47% | 1% | 1% | 118613 (10980) | 100% |

Table 3: Simulation results for Scenario 2(a) with 5 design features.

(A)

| Scenario 2(a) (5 designs) | Round 1 ORR[(1)] | Round 2 ORR[(2)] | Round 3 ORR[(3)] | Round 4 ORR[(4)] | Round 5 ORR[(5)] | Overall ORR | Adaptive Learning ORR | Highest true RR |
|---|---|---|---|---|---|---|---|---|
| Logistic ridge regression | 0.055 (0.000) | 0.067 (0.003) | 0.068 (0.001) | 0.068 (0.000) | 0.068 (0.000) | 0.066 (0.000) | 0.068 (0.000) | 0.068 (0.000) |
| Random forest | 0.055 (0.000) | 0.064 (0.004) | 0.067 (0.002) | 0.068 (0.001) | 0.068 (0.001) | 0.065 (0.002) | 0.068 (0.002) | 0.068 (0.000) |
| XGBoost | 0.055 (0.000) | 0.064 (0.003) | 0.067 (0.003) | 0.068 (0.001) | 0.068 (0.001) | 0.065 (0.001) | 0.068 (0.002) | 0.068 (0.000) |
| Ensemble learning - 3 learners | 0.055 (0.000) | 0.065 (0.004) | 0.068 (0.002) | 0.068 (0.001) | 0.068 (0.000) | 0.066 (0.000) | 0.068 (0.000) | 0.068 (0.000) |
| Ensemble learning - 7 learners | 0.055 (0.000) | 0.066 (0.003) | 0.068 (0.002) | 0.068 (0.000) | 0.068 (0.000) | 0.066 (0.000) | 0.068 (0.000) | 0.068 (0.000) |

(B)

| Scenario 2(a) (5 designs) | Round 1 Plan number | Round 2 Plan number | Round 3 Plan number | Round 4 Plan number | Round 5 Plan number | Expected rounds |
|---|---|---|---|---|---|---|
| Logistic ridge regression | 32 | 10.3 (6.0) | 4.1 (3.0) | 2.4 (1.4) | 1.6 (0.9) | 4.4 (0.7) |
| Random forest | 32 | 11.1 (7.8) | 4.6 (3.9) | 2.6 (2.6) | 1.8 (1.7) | 4.3 (0.8) |
| XGBoost | 32 | 11.2 (7.7) | 5.2 (4.5) | 3.1 (3.0) | 1.9 (1.7) | 4.4 (0.8) |
| Ensemble learning - 3 learners | 32 | 11.8 (7.7) | 4.1 (3.3) | 2.4 (2.1) | 1.9 (1.4) | 4.3 (0.7) |
| Ensemble learning - 7 learners | 32 | 10.9 (6.7) | 4.2 (3.4) | 2.3 (1.8) | 1.6 (1.3) | 4.3 (0.7) |

(C)

| Scenario 2(a) (5 designs) | Early Stopping | Due to plan number | Due to ORR restriction | Due to sample size | Sample size for last round | Better performance |
|---|---|---|---|---|---|---|
| Logistic ridge regression | 46% | 46% | 0% | 0% | 127187 (4652) | 100% |
| Random forest | 53% | 53% | 0% | 0% | 127869 (4859) | 98% |
| XGBoost | 47% | 44% | 3% | 0% | 127315 (4552) | 99% |
| Ensemble learning - 3 learners | 53% | 53% | 0% | 0% | 127449 (4581) | 100% |
| Ensemble learning - 7 learners | 56% | 56% | 0% | 0% | 127893 (4620) | 100% |

Table 4: Simulation results for Scenario 2(a) with 8 design features.

**(A)**

| Scenario 2(a) (8 designs) | Round 1 ORR$^{(1)}$ | Round 2 ORR$^{(2)}$ | Round 3 ORR$^{(3)}$ | Round 4 ORR$^{(4)}$ | Round 5 ORR$^{(5)}$ | Round 6 ORR$^{(6)}$ | Overall ORR | Adaptive Learning ORR | Highest true RR |
|---|---|---|---|---|---|---|---|---|---|
| Logistic ridge regression | 0.055 (0.000) | 0.064 (0.003) | 0.067 (0.002) | 0.068 (0.001) | 0.068 (0.000) | 0.068 (0.000) | 0.066 (0.000) | 0.068 (0.000) | 0.068 (0.000) |
| Random forest | 0.055 (0.000) | 0.059 (0.002) | 0.062 (0.003) | 0.065 (0.003) | 0.068 (0.002) | 0.068 (0.001) | 0.065 (0.002) | 0.067 (0.002) | 0.068 (0.000) |
| XGBoost | 0.055 (0.000) | 0.060 (0.002) | 0.063 (0.003) | 0.066 (0.002) | 0.068 (0.002) | 0.068 (0.001) | 0.066 (0.001) | 0.068 (0.001) | 0.068 (0.000) |
| Ensemble learning - 3 learners | 0.055 (0.000) | 0.061 (0.002) | 0.063 (0.003) | 0.066 (0.003) | 0.068 (0.002) | 0.068 (0.001) | 0.066 (0.001) | 0.068 (0.001) | 0.068 (0.000) |
| Ensemble learning - 7 learners | 0.055 (0.000) | 0.063 (0.002) | 0.064 (0.003) | 0.067 (0.002) | 0.068 (0.001) | 0.068 (0.000) | 0.066 (0.000) | 0.068 (0.000) | 0.068 (0.000) |

**(B)**

| Scenario 2(a) (8 designs) | Round 1 Plan number | Round 2 Plan number | Round 3 Plan number | Round 4 Plan number | Round 5 Plan number | Round 6 Plan number | Expected rounds |
|---|---|---|---|---|---|---|---|
| Logistic ridge regression | 256 | 106.0 (43.4) | 33.7 (31.7) | 11.1 (15.5) | 4.2 (5.1) | 2.5 (2.1) | 5.3 (0.8) |
| Random forest | 256 | 103.9 (57.2) | 33.8 (29.5) | 11.3 (14.3) | 3.9 (4.2) | 2.5 (2.1) | 5.5 (0.7) |
| XGBoost | 256 | 103.9 (51.6) | 38.7 (32.9) | 13.6 (17.5) | 4.3 (3.9) | 2.3 (1.6) | 5.6 (0.7) |
| Ensemble learning - 3 learners | 256 | 106.3 (53.9) | 37.2 (33.0) | 12.9 (16.9) | 5.2 (7.4) | 3.1 (3.7) | 5.5 (0.8) |
| Ensemble learning - 7 learners | 256 | 103.9 (44.4) | 34.5 (27.9) | 10.9 (12.0) | 4.3 (5.1) | 2.1 (1.7) | 5.6 (0.7) |

**(C)**

| Scenario 2(a) (8 designs) | Early Stopping | Due to plan number | Due to ORR restriction | Due to sample size | Sample size for last round | Better performance |
|---|---|---|---|---|---|---|
| Logistic ridge regression | 46% | 46% | 0% | 0% | 124938 (5455) | 100% |
| Random forest | 42% | 42% | 0% | 0% | 122630 (5002) | 98% |
| XGBoost | 31% | 30% | 1% | 0% | 122336 (4843) | 100% |
| Ensemble learning - 3 learners | 38% | 38% | 0% | 0% | 122846 (5679) | 100% |
| Ensemble learning - 7 learners | 31% | 31% | 0% | 0% | 122467 (5231) | 100% |

Table 5: Simulation results for Scenario 2(b) with 5 design features.

**(A)**

| Scenario 2(b) (5 designs) | Round 1 ORR$^{(1)}$ | Round 2 ORR$^{(2)}$ | Round 3 ORR$^{(3)}$ | Round 4 ORR$^{(4)}$ | Round 5 ORR$^{(5)}$ | Overall ORR | Adaptive Learning ORR | Highest true RR |
|---|---|---|---|---|---|---|---|---|
| Logistic ridge regression | 0.055 (0.000) | 0.061 (0.004) | 0.066 (0.005) | 0.068 (0.003) | 0.068 (0.002) | 0.064 (0.004) | 0.066 (0.005) | 0.068 (0.000) |
| Random forest | 0.055 (0.000) | 0.066 (0.003) | 0.068 (0.002) | 0.068 (0.001) | 0.068 (0.001) | 0.066 (0.001) | 0.068 (0.001) | 0.068 (0.000) |
| XGBoost | 0.055 (0.000) | 0.066 (0.002) | 0.068 (0.001) | 0.068 (0.001) | 0.068 (0.000) | 0.066 (0.001) | 0.068 (0.001) | 0.068 (0.000) |
| Ensemble learning - 3 learners | 0.055 (0.000) | 0.066 (0.003) | 0.068 (0.001) | 0.068 (0.001) | 0.068 (0.000) | 0.066 (0.002) | 0.068 (0.002) | 0.068 (0.000) |
| Ensemble learning - 7 learners | 0.055 (0.000) | 0.066 (0.002) | 0.068 (0.002) | 0.068 (0.001) | 0.068 (0.000) | 0.066 (0.001) | 0.068 (0.001) | 0.068 (0.000) |

**(B)**

| Scenario 2(b) (5 designs) | Round 1 Plan number | Round 2 Plan number | Round 3 Plan number | Round 4 Plan number | Round 5 Plan number | Expected rounds |
|---|---|---|---|---|---|---|
| Logistic ridge regression | 32 | 7.1 (3.9) | 2.9 (1.9) | 2.0 (1.3) | 1.6 (0.8) | 4.0 (0.9) |
| Random forest | 32 | 5.1 (4.0) | 2.9 (2.3) | 2.2 (1.6) | 1.5 (1.0) | 3.7 (1.0) |
| XGBoost | 32 | 4.8 (3.5) | 2.8 (2.0) | 2.0 (1.5) | 1.4 (0.8) | 3.7 (1.0) |
| Ensemble learning - 3 learners | 32 | 4.3 (3.2) | 2.3 (2.1) | 1.8 (1.5) | 1.6 (1.5) | 3.4 (0.9) |
| Ensemble learning - 7 learners | 32 | 5.1 (3.4) | 2.8 (1.9) | 1.9 (1.3) | 1.7 (0.8) | 3.8 (0.9) |

**(C)**

| Scenario 2(b) (5 designs) | Early Stopping | Due to plan number | Due to ORR restriction | Due to sample size | Sample size for last round | Better performance |
|---|---|---|---|---|---|---|
| Logistic ridge regression | 68% | 68% | 0% | 0% | 117944 (9500) | 84% |
| Random forest | 72% | 72% | 0% | 0% | 130706 (6259) | 98% |
| XGBoost | 77% | 72% | 2% | 0% | 130606 (6182) | 100% |
| Ensemble learning - 3 learners | 84% | 84% | 0% | 0% | 132461 (5840) | 98% |
| Ensemble learning - 7 learners | 74% | 74% | 0% | 0% | 129809 (5716) | 99% |

Table 6: Simulation results for Scenario 2(b) with 8 design features.

(A)

| Scenario 2(b) (8 designs) | Round 1 ORR[(1)] | Round 2 ORR[(2)] | Round 3 ORR[(3)] | Round 4 ORR[(4)] | Round 5 ORR[(5)] | Round 6 ORR[(6)] | Overall ORR | Adaptive Learning ORR | Highest ORR |
|---|---|---|---|---|---|---|---|---|---|
| Logistic ridge regression | 0.055 (0.000) | 0.061 (0.002) | 0.066 (0.003) | 0.068 (0.003) | 0.068 (0.002) | 0.068 (0.002) | 0.066 (0.002) | 0.068 (0.003) | 0.068 (0.000) |
| Random forest | 0.055 (0.000) | 0.060 (0.002) | 0.063 (0.003) | 0.066 (0.003) | 0.068 (0.001) | 0.068 (0.001) | 0.065 (0.002) | 0.067 (0.002) | 0.068 (0.000) |
| XGBoost | 0.055 (0.000) | 0.061 (0.003) | 0.064 (0.003) | 0.067 (0.003) | 0.068 (0.001) | 0.068 (0.001) | 0.066 (0.001) | 0.068 (0.001) | 0.068 (0.000) |
| Ensemble learning - 3 learners | 0.055 (0.000) | 0.061 (0.002) | 0.064 (0.003) | 0.067 (0.002) | 0.068 (0.002) | 0.068 (0.001) | 0.066 (0.001) | 0.068 (0.001) | 0.068 (0.000) |
| Ensemble learning - 7 learners | 0.055 (0.000) | 0.064 (0.003) | 0.065 (0.003) | 0.067 (0.003) | 0.068 (0.001) | 0.068 (0.001) | 0.066 (0.002) | 0.068 (0.002) | 0.068 (0.000) |

(B)

| Scenario 2(b) (8 designs) | Round 1 Plan number | Round 2 Plan number | Round 3 Plan number | Round 4 Plan number | Round 5 Plan number | Round 6 Plan number | Expected rounds |
|---|---|---|---|---|---|---|---|
| Logistic ridge regression | 256 | 91.9 (41.2) | 39.3 (31.4) | 15.4 (19.0) | 5.7 (7.7) | 2.5 (2.4) | 5.6 (0.8) |
| Random forest | 256 | 92.6 (55.2) | 30.5 (34.8) | 9.6 (13.3) | 3.6 (2.8) | 2.4 (1.7) | 5.4 (0.8) |
| XGBoost | 256 | 80.4 (58.6) | 27.3 (30.4) | 9.5 (13.0) | 3.7 (3.2) | 2.1 (1.6) | 5.2 (0.9) |
| Ensemble learning - 3 learners | 256 | 81.3 (53.0) | 25.3 (26.5) | 7.3 (10.2) | 3.0 (3.5) | 1.8 (1.2) | 5.2 (0.8) |
| Ensemble learning - 7 learners | 256 | 63.2 (42.8) | 20.8 (22.8) | 6.9 (8.8) | 3.4 (3.4) | 2.4 (2.1) | 5.2 (0.9) |

(C)

| Scenario 2(b) (8 designs) | Early Stopping | Due to plan number | Due to ORR restriction | Due to sample size | Sample size for last round | Better performance |
|---|---|---|---|---|---|---|
| Logistic ridge regression | 28% | 28% | 0% | 0% | 123963 (5386) | 96% |
| Random forest | 41% | 41% | 0% | 0% | 122710 (6585) | 98% |
| XGBoost | 53% | 53% | 0% | 0% | 123598 (6828) | 99% |
| Ensemble learning - 3 learners | 56% | 56% | 0% | 0% | 123357 (6430) | 100% |
| Ensemble learning - 7 learners | 54% | 54% | 0% | 0% | 123036 (6263) | 97% |