# OpenReview forum: "Enhancing Patient Recruitment Response in Clinical Trials: an Adaptive Learning Framework"
_auai.org/UAI/2024/Conference — UAI 2024 poster_

### Official Review · Reviewer_zWMm · 2024-03-17

**Q2-1 Originality-Novelty:** 2
**Q2-2 Correctness-Technical Quality:** 3
**Q2-5 Clarity Of Writing:** 3

**Q1 Summary And Contributions:**

The authors provide mathematically-rigorous framework for adaptive clinical trial recruitment. They propose to employ an ensemble learning paradigm where up to 7 different models are used to predict prediction rate to the recruitment. Several model settings are experimented on the simulation data.

**Q2-3 Extent To Which Claims Are Supported By Evidence:**

2: Fair: the main claims are somewhat supported by evidence (but the experimental evaluation may be weak, or does not match entirely with the claims, important baselines may be missing, proofs contain important ideas but lack rigor, algorithmic details are only discussed superficially, references are imprecise, assumptions are not sufficiently motivated or explicated, etc.).

**Q2-4 Reproducibility:**

3: Good: key resources (e.g. proofs, code, data) are available and key details (e.g. proofs, experimental setup) are sufficiently well-described for competent researchers to confidently reproduce the main results.

**Q3 Main Strengths:**

The details of mathematical construction for the framework are impressive.

The authors do a great job of explaining how and why the clinical trial recruitment can benefit from more systematic approaches.

**Q4 Main Weakness:**

As most of the readers will not be familiar with the previous works in this direction, it is required that authors provide a section on related works for better understanding.

The paper can benefit from more traditional-style ML paper organizations. For instance, it is hard to grasp what the evaluation metric is (Although It's ORR, more effort needs to be put into explicitly stating this in Section 4) and how each model of the ensemble model is trained (hyperparameter specification for random forest, what NN the authors are using, etc)

While I appreciate the mathematically-rigorous clinical trial recruitment framework the authors bring, I am not sure UAI is the best venue for this type of work. For instance, current conclusion and discussion are typically suited for journal-type submissions. I would like to suggest other more clinical journals for this purpose.

I was a bit surprised that f_k and g_k ended up being quite simple choices. Can authors provide what other choices exist in the field and include them in the paper? There needs to be ablation experiments around the choice of f_k and g_k

**Q5 Detailed Comments To The Authors:**

I think explanation on why only simulation data had to be used is lacking.

How does Section 5 fit into all this? Do you have experiments for this scenario?

**Q9 Complying With Reviewing Instructions:**

Yes

---

> ### Author Rebuttal · Authors · 2024-04-09
>
> We are grateful for reviewer’s insightful comments and constructive feedback. We have carefully deliberated and revised the manuscript to address the critiques, aiming to provide greater clarity and strengthen the methodological justifications:
> # Main Weakness:
> Q1. We add a Related Work section and details in the Appendix to demonstrate the relationship between the proposed approach and sequential adaptive designs:
> The adaptive learning approach shares similarities with sequential adaptive designs, but they have distinct focuses.
> 1. Study focus: Sequential adaptive designs have an “arm-oriented” focus, aiming to identify the best treatment (arm) among the tested treatments. In contrast, the adaptive learning methods in our trial recruitment have a “response-oriented” focus. It aims to improve the overall recruitment response rate until little improvement can be made, irrespective of which recruitment plan (arm) achieves a good response.
> 2. Scale of interventions: Traditional sequential adaptive designs can only handle a few treatments (small $K$) using classical statistical methods, whereas our approach is better suited for AI/ML techniques to systematically search within a large space of recruitment plans (large $K$).
> 3. Stage in clinical trials: Sequential adaptive designs are implemented to allocate treatments during the intervention stage, while it is often costly and takes a few years to run even with a limited sample size. In contrast, the proposed adaptive learning framework targets the recruitment stage, which is fast-paced with a huge sample space (e.g., 175,000 in the SilverSneakers study).
> Consequently, to achieve maximum efficiency, the methodologies tailored for these two types of studies differ. Our approach establishes an adaptive machine learning framework specifically designed to enhance patient recruitment in clinical trials.
>
> Q2. We have re-organized the last section and moved the discussion to the Extensions of the proposed framework section, including 1. sequential assignment of recruitment plans for non-responders, 2. subgroup learning, 3. refinement of clustering (ordering) methods, and 4. improvement with contextual MAB approach and Bayes learning.
> The updated conclusion section in the text is:
> “Patient recruitment remains challenging in large-scale pragmatic clinical trials. Traditional strategies often struggle to meet demanding recruitment requirements. We proposed a novel adaptive learning framework integrating ensemble learning to iteratively optimize patient recruitment. Through simulations, we demonstrated our framework's ability to efficiently identify and prioritize the most effective recruitment plans while mitigating the risk of selecting suboptimal recruitment plans. Our adaptive framework represents a significant method advancement with the potential to substantially improve recruitment rates by facilitating more efficient pragmatic trials. While further evaluation is needed, this work establishes a foundation for leveraging ML to address recruitment challenges in clinical studies.”
> Hope you will find it more suitable for UAI.
>
> Q3. We thank the reviewer for this insightful point. In this framework paper we highlight the improvement only using the simple $f$ and $g$ functions, while larger improvement is expected with careful calibration. We will investigate this problem by comparing different $f$s and $g$s and give recommendations in future studies.
> # Detailed Comments
> Q1.
> * The simulation study demonstrates how effective this approach could be compared with existing recruitment strategy.
> * Simulation results will be helpful to determine the choice of design parameters. E.g., through simulation we usually see for T>10, the benefit of adaptive learning with additional cohort will be very limited although at the cost of increasing the total recruitment time will be prolonged. This will help develop a practical design of patient recruitment plan for real trials.
> Q2.
> We have added an explanation of the original Section 5 to demonstrate the difference between the proposed method and the sequential assignment for non-responders in Section 5:
> “The proposed adaptive learning approach is designed to sequentially assign recruitment plans to potential patients. The entire pool of eligible patients will be divided into $K$ cohorts. Each patient within a cohort will be assigned a single recruitment plan. However, in the context of rare diseases where the pool is limited, non-responders may be re-assigned to receive additional recruitment plans. This approach allows for multiple recruitment strategies to be utilized for patients who do not initially respond, maximizing the chances of successful enrollment in the study.” The context of sequential assignment for non-responders is not restricted to traditional clinical trials. It can be further applied to enhance survey responses in social/political studies.
> We don’t have experiments for this scenario.

---

### Official Review · Reviewer_mD5T · 2024-03-18

**Q2-1 Originality-Novelty:** 2
**Q2-2 Correctness-Technical Quality:** 3
**Q2-5 Clarity Of Writing:** 3

**Q10 Ethical Concerns:**

No.

**Q1 Summary And Contributions:**

This paper addresses the critical challenge of patient recruitment in clinical trials, and low response rate in recruitment
process. It highlights the specific challenges faced in pragmatic trials, emphasizing the need for large sample size and the necessity of efficient patient recruitment. While existing solutions have limitations, AI and ML offer promise in participant identification and selection. However, challenges remain in encouraging participant responses. The paper proposes leveraging ensemble learning to develop an adaptive recruitment strategy, aiming to enhance response rate by iteratively refining the selection of effective recruitment plans. This approach aims to revolutionize clinical trial recruitment and improve generalizability by optimizing recruitment plan allocation.

**Q2-3 Extent To Which Claims Are Supported By Evidence:**

3: Good: the main claims are supported by convincing evidence (in the form of adequate experimental evaluation, proofs, (pseudo-)code, references, assumptions).

**Q2-4 Reproducibility:**

2: Fair: key resources (e.g. proofs, code, data) are unavailable but key details (e.g. proof sketches, experimental setup) are sufficiently well-described for an expert to confidently reproduce the main results.

**Q3 Main Strengths:**

1. The paper introduces an adaptive learning procedure integrated with ensemble learning to address the challenge of patient recruitment in clinical trials, particularly in pragmatic trials requiring large sample size.
2. The efficacy of the proposed approach is validated through a simulation study.
3. The approach is tailored to address challenges inherent in rare disease settings as well.
4. The proposed method and the experimental settings seem technically sound to me.

**Q4 Main Weakness:**

1. The codes are not available so it was not possible to run the experiments and reproduce the results.
 2. The proposed approach involves sophisticated techniques such as ensemble learning and clustering, which may introduce complexity and computational burden, potentially hindering its practical implementation.
3. Also to what extent this model can be generalized to other clinical settings.
4. Lack of the proof of lemma.

**Q5 Detailed Comments To The Authors:**

1. How choice of f function (pre-specified randomization rule) could affect the results?
2. How you would compute the the predicted response rate, did you use the real data for that?
3. You also fixed the maximum rounds at 6, how it improves/does not improve the results if you let the data estimates it?
4. Was there any computational issues in running the models?
5. What would you think about adding the feature selection to the proposed model?
6. The derivation of Lemma 1 was not clear to me, maybe I am missing something, could you please clarify this or even adding the proofs to the appendix?
7. Are there any assumptions which need to be held in real data?
8. It might be good to explain briefly the algorithm one in the text.
9. What do the values in parenthesis indicate in table 1?
10. Please make sure to use the complete terminology upon its initial introduction; for instance, when AI and ML are first mentioned in the text, use "Artificial Intelligence (AI)" and "Machine Learning (ML)" instead, use the abbreviations for subsequent references.
11. There are some typo-errors in the text: page 5: don't, page 3: iff; false discovery control or false discovery rate? page 4: below lemma 2, definition of g(k). Page 6: (0.097 (0.003)): extra parenthesis, page 1: indivituals
12. You used upper case for Round and also lower case, it is good to keep it consistent in the entire text. Same for the "cohort".

**Q9 Complying With Reviewing Instructions:**

Yes

---

> ### Author Rebuttal · Authors · 2024-04-09
>
> We are grateful for reviewer’s insightful comments and constructive feedback. We have carefully deliberated and revised the manuscript to address the critiques:
>
> # Main Weakness:
> Q1. The reproducible code is provided on [GitHub](https://github.com/vivid225/Integrating-Machine-Learning-Methods-in-Clinical-Trial-Recruitment).
>
> Q2.
> 1. Proposed framework is flexible. To reduce the complexity, any single ML method encapsulated in ensemble learning can be applied as a standalone approach in adaptive learning process:
> •	Theoretically, Lemmas 1 and 2 still hold, to guarantee the properties of non-inferiority, optimality, and consistency.
> •	As illustrated in the simulation study (e.g. Table 1), single ML methods, though not as effective as ensemble model, still yield profound recruitment rate improvements.
> 2. Clustering is not mandated. Our framework only requires a ranking method on recruitment plans for the pruning process, and the adaptive learning will select plans with good performance for future cohorts of patients.
> 3. Overall, we believe the recruitment benefits outweigh the implementation challenges.
> 4. The primary goal of this paper is to introduce a novel adaptive learning framework for enhancing recruitment responses. To facilitate implementation, we plan to develop open-source user-friendly tools, as we usually do for our methods.
>
> Q3. The proposed framework is general-purpose (see responses to Q7 below). Additionally, we add a section in manuscript to discuss promising model generalization.
>
> Q4. The proofs of Lemma 1 and Lemma2 have been included in the Supplementary Material.
>
> # Detailed Comments
> Q1. In this framework paper we try to highlight the improvement only using the simple $f$ and $g$ functions, while larger improvement is expected with careful calibration. We will investigate this problem by comparing different $f$s and $g$s and give recommendations in future studies.
>
> Q2. We use real data to compute the predicted response rate at round $t$.
>
> We add a clarification for Algorithm 1:
> ``In the initial round ($t=1$), all treatment plans are assigned equal probabilities $w_k^{(1)} = 1/K, k = 1,\cdots, K$, and patient responses are collected as $D^{(1)}$. For subsequent rounds $t > 1$, we follow the steps of Figure 1 to perform the adaptive allocation. Specifically, at round $t$, we have response data $D^{t-1}$ from previous rounds. Given $D^{t-1}$, an (ensemble) learning model is conducted to predict the response of recruitment plans $\hat{\boldsymbol{p}}^{(t-1)}$. With the predicted response rates $\hat{\boldsymbol{p}}^{(t-1)}$, we can calculate the allocation rates $\boldsymbol{W}^{(t)}$ and randomly assign the recruitment plans to patients in Cohort $t$ based on the allocation rates. Data $D^{t}$ are thus collected after patients respond to the assigned plans. This iterative process continues until the maximum number of rounds $T$ is reached or one of the early termination conditions 3(a)-(b) in Algorithm 1 is met.’’
>
> Q3. We add the following practical guideline in the manuscript:
> ``In practice, researchers should begin with a conservative estimate of the time required for each round, denoted as $Time_r$ (e.g., 2 months). Given the total scheduled duration, $Tot_t$ (e.g., 1 year), the maximum number of rounds $T_{max}$ can be determined as $T_{max} = Tot_t / Time_r$. The total recruitment cost should then be evaluated for $T_{max}$ rounds to meet the cost constraints.
>
> For effective adaptive learning, we recommend $T=4$ to $8$ rounds, which tends to strike a balance between computational efficiency and the ability to learn and adapt over multiple iterations. Specific choice of $T$ can be validated with simulation study tailored to the trial context. The final selection should be made jointly with the study's principal investigator, considering the overall recruitment strategy and resource limitations.’’
>
> Q4. There reproducible code is available on [GitHub](https://github.com/vivid225/Integrating-Machine-Learning-Methods-in-Clinical-Trial-Recruitment). There was no specific computational issue in running the model since we used the established Python packages for the machine learning models.
>
> Q5. We add a discussion of the possible model extensions including feature selection. We think it’s a great idea.
>
> Q6. See main weaknesses Q4.
>
> Q7. In this paper we tried to make the proposed framework as general as possible, so there is no additional assumption to be held except for the non-inferiority and optimality/consistency properties specified in the lemmas. In practice, we assume the patient sampling to be random. Otherwise, this approach is general-purpose.
>
> Q8. See the answer for detailed comments Q2.
>
> Q9. Standard deviation. It is also explained in the table caption.
>
> Q10-Q12. Typo issues have been fixed in the text. For **iff**, it is a mathematical abbreviation for **if and only if**.

---

### Official Review · Reviewer_UwAS · 2024-03-20

**Q2-1 Originality-Novelty:** 2
**Q2-2 Correctness-Technical Quality:** 2
**Q2-5 Clarity Of Writing:** 3

**Q1 Summary And Contributions:**

The paper introduces an adaptive learning procedure to improve patient recruitment for medical trials. The procedure uses ensemble learning to predict the (average) response rate of different recruitment plans and uses adaptive learning in a sequential setting to drop recruitment plans with low response rate early. They compare their learning procedure in different simulations to a random baseline and show the robustness of their ensemble learner.

**Q2-3 Extent To Which Claims Are Supported By Evidence:**

3: Good: the main claims are supported by convincing evidence (in the form of adequate experimental evaluation, proofs, (pseudo-)code, references, assumptions).

**Q2-4 Reproducibility:**

2: Fair: key resources (e.g. proofs, code, data) are unavailable but key details (e.g. proof sketches, experimental setup) are sufficiently well-described for an expert to confidently reproduce the main results.

**Q3 Main Strengths:**

The paper is well structured and the application setting is very interesting. I am not very familiar with the adaptive learning literature, so I cannot say if the presented procedure is novel, but their contributions are presented convincingly.

**Q4 Main Weakness:**

- I would have appreciated a related work section (even if just in the appendix). The problem seems conceptually similar to existing problems of choosing a "treatment" for a group of people (e.g., A/B testing in advertising), but there was no discussion of related  methods, why these could or could not be applied etc. Hence, also the only baseline compared to is a uniform sampling baseline.
- Section 4 was hard to follow. In particular, the different experiment scenarios and the benchmark/baseline were unclear, it would be better to move some of the information in the Appendix to the main.
- Some parts of the procedure were not clear (see questions below)

**Q5 Detailed Comments To The Authors:**

- How do you guarantee that the pruning factor of the clustering procedure is monotone? Do you just drop the recruitment plans that were pruned in the previous round from the clustering? This was not very clear in the text.
- From the paper, I understand that a semi-strict pruning factor is only defined by the pruning behavior in the final round $T$. Hence, it could prune a recruitment plan $k$ ($k> K_0$) with higher response rate in an earlier round than a recruitment plan $k'$ with lower response rate ($k'> k$). However, in contrast to what is stated in the paper, that could lead to a pruning sequence that does not satisfy $\sum_k w_k^{(t)} p_k \geq \sum_k w_k^{(t-1)} p_k$, no?

**Typos and minor questions:**
- page 4 Lemma 2: $k \neq 1p_k$ should be $k \neq 1$?
- page 5 Definition $n_{min}^{(t)}$: what are $Z_{1-\alpha}$ and  $Z_{1-\beta}$, are those values from the z table of a normal distribution?
- page 5 subsection 3.3.3: it is stated that $\widehat{ORR}^{(t)}$ is computed with $D^{(t)}$, but shouldn't both $\widehat{ORR}^{(t)}$ and $\widehat{ORR}^{(t-1)}$ be computed with $D^{(t-1)}$ as $D^{(t)}$ is not available yet?
- page 6 top left: Notation $Time_r$ and $Time_n$ are used, I guess the later is a typo?

**Q9 Complying With Reviewing Instructions:**

Yes

---

> ### Author Rebuttal · Authors · 2024-04-09
>
> We are grateful for reviewer’s insightful comments and constructive feedback. We have carefully deliberated and revised the manuscript to address the critiques, aiming to provide greater clarity and strengthen the methodological justifications:
>
> # Main weaknesses
> Q1. We thank the reviewer for the great comment, and agree that this work could be conceptually confused with existing sequential learning problems. However, we’d like to first justify the **primary contribution** of this paper is to propose a novel framework for optimizing trial recruitment. To the best of our knowledge, this work represents the first application of an adaptive, data-driven approach to enhancing recruitment response rates in clinical trials.
>
> In the supplementary materials of the manuscript, we have added a related work section about the related work in traditional *sequential adaptive designs* and discussion of several key differences in study focus, scale of the problem, and practical considerations.
>
> To illustrate this groundbreaking framework, we adapt and tailor the ensemble learning and clustering for the trial recruitment problem to illustrate the main idea. We restructure section 5 to discuss its promising extensions including 1. sequential assignment of recruitment plans for non-responders, 2. subgroup learning, 3. refinement of clustering (ordering) methods, and 4. improvement with contextual MAB approach and Bayes learning. Specifically, we have included the following paragraph:
> "While ensemble learning offers a powerful approach to identifying superior recruitment plans, its focus solely on maximizing predicted plan response rates might miss broader optimization opportunities. Alternative methods, such as the contextual multi-armed bandit approach [Oh and Iyengar, 2019], which simultaneously balances exploration (trying new recruitment plans) with exploitation (focusing on successful ones), may lead to a more efficient adaptive recruitment strategy by globally maximizing the total predictive reward. Additionally, when relevant external data or experts' opinions are available, the allocation optimization could also be conducted using the Bayes learning [Gelman et al., 2013]. Moreover, in clinically diverse settings with multiple patient groups, it is compelling to employ learning methods that can incorporate both plan features and patient characteristics. Potentially, it will lead to better identification of optimal recruitment plans for specific patient subgroups, opening a door to diverse efficiency gains."
>
> Q2.
> Due to the lack of data-driven methods in trial recruitment literature, we have clarified in the manuscript that the benchmark is the traditional one-stage recruitment strategy to randomly assign participants to various recruitment plans, and tested them in different scales of problem complexity (simple with 5 features for 32 designs vs difficulty with 8 features for 256 designs).
> Additionally, we have added an Algorithm 2 in Supp. Mat., to specify how the procedure is conducted with clustering in the simulation study (Algorithm 1 is generic).
>
> # Detailed comments:
> Q1.
> We have added the following detail in the text to clarify the functionality of the pruning factor:
> "we assign the adaptive pruning factor $g_k^{(t)}$ at value 1 to a cluster of recruitment plans with the highest predicted response rates in round $t$, denoted by $C^{(t)}$.  All the recruitment plan $k \notin C^{(t)}$ are pruned with $g_k^{(t)}=0$. By applying the K-means method, this selected recruitment plan set $C^{(t)}$ is determined out of the previously selected $C^{(t-1)}$ in round $t-1$ to satisfy the monotonic condition. The silhouette score is used to select the best number of clusters. If it demonstrates the effectiveness of recruitment in this simple setting, we expect the adaptive learning performance to be further enhanced with tailored ML methods in real data applications."
> It is a nested clustering procedure. The clustering procedure at round t is only applied to preselected set $C^{(t-1)}$ from round $t-1$. If $g_k^{(t-1)}=0$, $g_k^{(t)}$ will also be 0. The pruning factor consistently exclude the less effective recruitment plans among the preceding round, which impose the monotonic condition.
>
> Q2. As discussed in the previous question, the adaptive plan selection is a nested pruning process and the monotonic condition is satisfied.
> Considering $\sum_k w_k^{(t)} p_k = \sum_k f_k^{(t)} g_k^{(t)} p_k / (\sum_k f_k^{(t)} g_k^{(t)}) $ and $\sum_k w_k^{(t-1)} p_k = \sum_k f_k^{(t-1)} g_k^{(t-1)} p_k / (\sum_k f_k^{(t-1)} g_k^{(t-1)}) $, the pruning factor selects more effective plans at round $t$ out of plans from round $t-1$, so the inequality is naturally satisfied.
>
> Q3. Typos and minor questions:
> * Typos are fixed in the text.
> * We add the explanation in the text that “$Z_{1-\alpha}$ and $Z_{1-\beta^{(t)}}$ are critical values from the standard normal distribution.”

---

### Official Review · Reviewer_xxXi · 2024-03-22

**Q2-1 Originality-Novelty:** 2
**Q2-2 Correctness-Technical Quality:** 4
**Q2-5 Clarity Of Writing:** 3

**Q1 Summary And Contributions:**

The authors study the problem of adaptive clinical trial recruitment. Given several possible interventions, and several subpopulations, they wish to allocate the interventions to sequentially optimize the positive outcome. Early stopping is permitted if power permits.

**Q2-3 Extent To Which Claims Are Supported By Evidence:**

2: Fair: the main claims are somewhat supported by evidence (but the experimental evaluation may be weak, or does not match entirely with the claims, important baselines may be missing, proofs contain important ideas but lack rigor, algorithmic details are only discussed superficially, references are imprecise, assumptions are not sufficiently motivated or explicated, etc.).

**Q2-4 Reproducibility:**

3: Good: key resources (e.g. proofs, code, data) are available and key details (e.g. proofs, experimental setup) are sufficiently well-described for competent researchers to confidently reproduce the main results.

**Q3 Main Strengths:**

The paper is well written and clear to read, and all the statements appear to be correct as far as I can tell.

**Q4 Main Weakness:**

The paper fails to demonstrate how it is novel over the state of the art. Furthermore the paper omits full details of proofs and leaves it to the reader.

**Q5 Detailed Comments To The Authors:**

My main critique is that while the paper is good, it is not at all clear what constitutes the novel contribution. I am personally aware of work in statistics such as sequential adaptive designs, which study very similar settings (multiple treatments, multiple stages, where some utility function is to be optimized) and these works are fairly old at this point.

Moreover, the authors do not provide a related work section that shows how this work (specifically the adaptive recruitment part) relates to the broader literature.

Here are some works that I found in my cursory search: I would appreciate a commentary on at least these works, if not others that you must have come across studying this topic.

F. Hu and W. F. Rosen-
berger. The theory of response-adaptive random-
ization in clinical trials, volume 525. John Wiley
& Sons, 2006

Theodore G. Karrison, Dezheng Huo, Rick Chappell,
A group sequential, response-adaptive design for randomized clinical trials,
Controlled Clinical Trials,
Volume 24, Issue 5,
2003,
Pages 506-522,

Li, Zhengqing, et al. “An Adaptive Design for Maximization of a Contingent Binary Response.” Lecture Notes-Monograph Series, vol. 25, 1995, pp. 179–96. JSTOR, http://www.jstor.org/stable/4355843. Accessed 22 Mar. 2024.

**Q9 Complying With Reviewing Instructions:**

Yes

---

> ### Author Rebuttal · Authors · 2024-04-09
>
> We are grateful to the reviewers for their insightful comments and constructive feedback on our original manuscript. We have carefully deliberated each point raised and substantively revised the manuscript to address the critiques, aiming to provide greater clarity and strengthen the methodological justifications. We believe these revisions have significantly strengthened the proposal. A summary of major revisions is provided below:
>
> # Main Weakness
> Q1. We thank the reviewer to raise this critical point and we politely disagree.
>
> The primary focus of this paper is to propose a novel framework for optimizing trial recruitment. To the best of our knowledge, this work represents the groundbreaking application of an adaptive, data-driven approach to enhancing recruitment response rates in clinical trials.
>
> The proposed framework differs from sequential adaptive designs in several key ways. The response to *Detailed Comments* below provides further details.
>
>
> Q2. The proofs of Lemma 1 and Lemma2 have been included in the supplementary material.
> "
> Lemma 1:
> According to the Cauchy-Schwarz inequality, we can get
> $$(\sum_k f_k * 1)^2 \leq K\sum_k f_k^2$$
> The inequality is rearranged to
> $$\sum_k f_k^2 / \sum_k f_k \geq \sum_k f_k /K$$
> Thus, with $f_k(p_1, p_2, \cdots, p_K) \propto p_k$, we can get
> $$\sum_k f_kp_k / \sum_k f_k \geq \sum p_k / K.$$
>
> Lemma 2:
> or $1 \geq p_1 \geq p_2 \geq \cdots>p_K \geq 0$, we have
> \begin{eqnarray*}
>     \sum_k w_k^{(t)} p_k &=& \sum_k f_k g_k^{(t)} p_k / \sum_k f_k g_k^{(t)} \\
>     &\leq& \sum_k f_k g_k^{(t)} p_1 / \sum_k f_k g_k^{(t)} = p_1
> \end{eqnarray*}
> Also, we know that $g_k^{(T)} = 1$ if $k = 1$, so
> \begin{eqnarray*}
>     \sum_k f_k g_k^{(T)} p_k / \sum_k f_k g_k^{(T)}= f_1 p_1 / f_1 = p_1.
> \end{eqnarray*}
> Therefore, we can conclude that
> \begin{eqnarray*}
>     \sum_k w_k^{(t)} p_k &=& \sum_k f_k g_k^{(t)} p_k / \sum_k f_k g_k^{(t)} \\
>     &\leq& \sum_k f_k g_k^{(T)} p_k / \sum_k f_k g_k^{(T)}= p_1.
> \end{eqnarray*}"
>
> # Detailed Comments
> Q1. We have added the following clarification in the manuscript and supplementary material:
>
> "The proposed adaptive learning approach shares similarities with sequential adaptive designs, which dynamically update the allocation ratios during the trial to minimize the number of patients assigned to inferior treatments while preserving randomization and statistical power.
> However, they are distinct in the following key aspects:
> * Study focus:
> Sequential adaptive designs have an "arm-oriented" focus, aiming to identify the best treatment (arm) among the tested treatments. In contrast, the adaptive learning methods in our trial recruitment have a ``response-oriented'' focus. It aims to improve the overall recruitment response rate until little improvement can be made, irrespective of which recruitment plan (arm) achieves a good response.
> * Scale of interventions: Traditional sequential adaptive designs can only handle a few treatments (small $K$) using classical statistical methods, whereas our approach is better suited for AI/ML techniques to systematically search within a large space of recruitment plans (large $K$).
> * Stage in clinical trials: Sequential adaptive designs are implemented to allocate treatments during the intervention stage, while it is often costly and takes a few years to run even with a limited sample size. In contrast, the proposed adaptive learning framework targets the recruitment stage, which is fast-paced with a huge sample space (e.g., 175,000 in the SilverSneakers study).
>
> Consequently, the design methodologies needed for these two types of studies differ significantly, and ML methods are naturally suited in the recruitment setting to optimize its efficiency. Our proposed approach establishes an adaptive ML framework to enhance participant recruitment in clinical trials. To the best of our knowledge, this is a groundbreaking development in this field.’’

---

### Official Review · Reviewer_ur4E · 2024-03-22

**Q2-1 Originality-Novelty:** 2
**Q2-2 Correctness-Technical Quality:** 3
**Q2-5 Clarity Of Writing:** 3

**Q1 Summary And Contributions:**

This paper suggests a method of optimally choosing the recruitment plan for clinical trials using the adaptive learning method.

**Q2-3 Extent To Which Claims Are Supported By Evidence:**

2: Fair: the main claims are somewhat supported by evidence (but the experimental evaluation may be weak, or does not match entirely with the claims, important baselines may be missing, proofs contain important ideas but lack rigor, algorithmic details are only discussed superficially, references are imprecise, assumptions are not sufficiently motivated or explicated, etc.).

**Q2-4 Reproducibility:**

3: Good: key resources (e.g. proofs, code, data) are available and key details (e.g. proofs, experimental setup) are sufficiently well-described for competent researchers to confidently reproduce the main results.

**Q3 Main Strengths:**

1. I appreciate the paper's approach to engaging readers. I agree that the issue addressed in the paper is important. The motivation is effectively supported by real-world examples.
2. The authors provide the simple solution for an important problem.

**Q4 Main Weakness:**

1. The paper lacks mathematical formalization. For example, I think the explanation on how the clustering has been done with which samples is insufficient. Also, I don’t see any reasoning on the monotonic condition imposed on the g function.
2. I don’t see the difference between Figure 2 and 3. In other words, I felt that Section 5 is redundant.
3. I felt that this problem can be considered in an RL framework (Multi-armed Bandit or Markov Decision Process). Using these tools, we can rank which arms are more beneficial. I think discussing the potential of RL frameworks is beneficial.

**Q5 Detailed Comments To The Authors:**

1. “We are expecting the high assignment probabilities w(t) for effective recruitment plans (i.e., recruitment plan k’s with high response rate pk) will increase over time, while an decrease or even be reduced to 0 for non-competitive recruitment plans (i.e., recruitment plan k’s with low response rate pk).” is grammatically flawed.
2. “For the adaptive pruning factor g^(t)_k, we apply the K-means method [Lloyd, 1982, MacQueen, 1967], along with the silhouette score [Rousseeuw, 1987] to select the best number of clusters, to cluster on the predicted response rates and use g^(t)_k to label the cluster C(t) with the highest response rates” — I don’t get it. What are the samples that you are clustering?
3. I don’t see the difference between Figure 2 and 3. Aren’t they equivalent strategy?
4. Any practical guideline for choosing $T$?
5. Have you considered an individualized recruitment strategy using the RL framework (e.g., contextual Bandit).

**Q9 Complying With Reviewing Instructions:**

Yes

---

> ### Author Rebuttal · Authors · 2024-04-09
>
> We are grateful for reviewer’s insightful comments and constructive feedback. We have carefully deliberated and revised the manuscript to address the critiques, aiming to provide greater clarity and strengthen the methodological justifications:
>
> # Main Mistakes:
> Q1:
> * We add proofs of lemmas in Supp. Mat.
> * We add an Algorithm 2 in Supp. Mat., to specify how the procedure is conducted with clustering in the simulation study (Algorithm 1 is generic):
> "
> 1. Given the data $D^{(t-1)}$ collected up to round $t-1$,
>     - Apply a learning model to predict the plan response rates $\hat{p}_k^{(t-1)}$ among the recruitment plans $k\in C^{(t-1)}$ (i.e., the set of recruitment plans with the adaptive pruning factor $g^{(t-1)}_k=1$);
>     - Perform K-means clustering on the predicted plan response rates $\{\hat{p}_k^{(t-1)}\}$, $k\in C^{(t-1)}$;
>     - Assign (keep) $g^{(t)}_k=1$ to the recruitment plans in the best-performed cluster, denoted by $C^{(t)}$. All the other recruitment plans $k \notin C^{(t)}$ are pruned with $g^{(t)}_k = 0$;
> 2. Calculate the allocation rates $\boldsymbol{W}^{(t)} = (w_1^{(t)}, w_2^{(t)}, ..., w_K^{(t)})$ with
>       $$w_k^{(t)} = \frac{\hat{p}_k^{(t-1)} \cdot g_k^{(t)} }{\sum_k \hat{p}_k^{t-1} \cdot g_k^{(t)} }$$
> "
>
> * For monotonicity of pruning factor $g$: We clarify on the nested clustering procedure. "The clustering at round t is only applied to preselected set $C^{(t-1)}$ from round $t-1$. If $g_k^{(t-1)}=0$, $g_k^{(t)}$ will also be 0. The pruning factor consistently excludes the less effective recruitment plans among the preceding round, which impose the monotonic condition."
>
> Q2: In the original adaptive learning framework (Fig.2), we divide all patients into $K$ disjoint cohorts (rounds). Each patient is assigned only once, to a single recruitment plan. However, in rare diseases where the pool of patients is limited (Fig.3), all patients are assigned in round 1, randomly to a single recruitment plan; non-responders will be continually re-assigned, to other recruitment plans learned to be effective in previous rounds. This approach allows for recruitment strategies for patients who do not initially respond, maximizing the chances of successful enrollment.
>
> Q3: We restructure section 5 to discuss promising extensions including 1. sequential assignment of recruitment plans for non-responders, 2. subgroup learning, 3. refinement of clustering (ordering) methods, and 4. improvement with contextual MAB approach and Bayes learning:
> "$\dots$ Alternative methods, such as the contextual multi-armed bandit approach, which simultaneously balances exploration (trying new recruitment plans) with exploitation (focusing on successful ones), may naturally lead to a more efficient adaptive recruitment strategy by globally maximizing the total predictive reward. $\dots$"
>
> # Detailed comments:
> Q1: We rephrase the sentence:
> ``If the adaptive learning approach is effective, we expect the assignment probabilities $w_k^{(t)}$ for high-performing recruitment plans to increase over time. Conversely, the probabilities should decrease, potentially reaching zero, for underperforming recruitment plans with low response rates $p_k$. Overall, the algorithm will dynamically prioritize the more successful strategies while phasing out ineffective ones.’’
>
> Q2: We add the following clarification:
> "we assign the adaptive pruning factor $g_k^{(t)}$ at value 1 to a cluster of recruitment plans with the highest predicted response rates in round $t$, denoted by $C^{(t)}$.  All the recruitment plan $k \notin C^{(t)}$ are pruned with $g_k^{(t)}=0$. By applying the K-means method, this selected recruitment plan set $C^{(t)}$ is determined out of the previously selected $C^{(t-1)}$ in round $t-1$ to satisfy the monotonic condition. The silhouette score is used to select the best number of clusters. If it demonstrates the effectiveness of recruitment in this simple setting, we expect the adaptive learning performance to be further enhanced with tailored ML methods in real data applications".
>
> Q3: See main weaknesses, Q2.
>
> Q4: We add the following practical guidelines:
> ``In practice, researchers should begin with a conservative estimate of the time required for each round, denoted as $Time_r$ (e.g., 2 months). Given the total scheduled duration, $Tot_t$ (e.g., 1 year), the maximum number of rounds $T_{max}$ can be determined as $T_{max} = Tot_t / Time_r$. The total recruitment cost should then be evaluated for $T_{max}$ rounds to meet the cost constraints.
>
> For effective adaptive learning, we recommend $T=4$ to $8$ rounds, which tends to strike a balance between computational efficiency and the ability to learn and adapt over multiple iterations. Specific choice of $T$ can be validated with simulation study tailored to the trial context. The final selection should be made jointly with the study's principal investigator, considering the overall recruitment strategy and resource limitations.’’
>
> Q5: See main weaknesses, Q3.

---

### Meta-Review · Area_Chair_QJUv · 2024-04-18

The paper addresses an acute problem which has not seen much treatment by the ML community so-far: recruitment to clinical trials. The presented approach is novel and well-grounded in the practical concerns of running clinical trials, while addressing crucial statistical considerations. While the overall analysis could be made more rigorous (as pointed by several reviewers), it still merits publication; I hope this will encourage more work on this understudied yet highly impactful problem.
I ask the authors to make sure to include in the updated version of the paper the many important points brought up during the discussion phase.